# The Effect of β-Glucans from Oats and Yeasts on the Dynamics of Ice Crystal Growth in Acidophilic Ice Cream Based on Liquid Hydrolyzed Whey Concentrate

**DOI:** 10.3390/foods14132184

**Published:** 2025-06-22

**Authors:** Artur Mykhalevych, Galyna Polishchuk, Agata Znamirowska-Piotrowska, Anna Kamińska-Dwórznicka, Maciej Kluz, Magdalena Buniowska-Olejnik

**Affiliations:** 1Problem Scientific and Research Laboratory, National University of Food Technologies, Volodymyrska 68 St., 01033 Kyiv, Ukraine; artur0707@ukr.net; 2Department of Milk and Dairy Products Technology, Educational and Scientific Institute of Food Technologies, National University of Food Technologies, Volodymyrska 68 St., 01033 Kyiv, Ukraine; milknuft@i.ua; 3Department of Dairy Technology, Institute of Food Tehnology and Nutrition, University of Rzeszow, Ćwiklinskiej 2D St., 35-601 Rzeszow, Poland; aznamirowska@ur.edu.pl; 4Department of Food Engineering and Process Management, Institute of Food Sciences, Warsaw University of Life Sciences (WULS-SGGW), Nowoursynowska 159C, 02-776 Warsaw, Poland; anna_kaminska1@sggw.edu.pl; 5Andrzej Frycz Modrzewski Krakow University, Gustawa Herlinga-Grudzińskiego 1, 30-705 Krakow, Poland; mkluz@uafm.edu.pl

**Keywords:** ice cream, liquid whey concentrate, freezing point, ice crystals, quality indicators, oat β-glucan, yeast β-glucan

## Abstract

Improving the texture and shelf-life of whey-based ice cream remains a key challenge in clean-label food formulation. This study investigated the effects of different stabilizing ingredients—including Cremodan SI 320 (0.6%), oat β-glucan (0.25–0.5%), and yeast β-glucan (0.25–0.5%)—on the physicochemical properties and freezing dynamics of ice cream made from liquid hydrolyzed demineralized whey concentrate. Compared to Cremodan, oat β-glucan significantly lowered the freezing point, improved overrun, and enhanced melting resistance. Yeast β-glucan led to the smallest initial ice crystals (8.49 ± 0.37 μm) and minimal growth after one month (9.52 ± 0.16 μm), outperforming the control and Cremodan samples in crystal stability. The chemical composition and textural properties of each formulation were also evaluated. These findings demonstrate that oat and yeast β-glucans function as natural stabilizers, offering clean-label potential and improved structural integrity in frozen dairy desserts.

## 1. Introduction

Ice cream is a complex colloidal system, the texture and stability of which are influenced by various factors, including the presence of structural stabilizers (proteins and polysaccharides) that can affect the freezing point and prevent the growth of ice crystals during low-temperature processing and long-term storage [1,2]. Whey ice cream, made from fresh whey or its processed products, has a chemical composition that is significantly different from the traditional types of this product (10–16% fat). The high content of lactose and free water, low content of solids, especially fat and protein, results in increased susceptibility to structural defects such as ice recrystallization and loss of creaminess during storage. These weaknesses are particularly pronounced in acidophilic whey-based systems, which require tailored stabilization strategies to maintain product quality. In previous research, a formulation of whey-based ice cream using liquid concentrates of demineralized whey was developed, and its interaction with whey protein isolate was investigated. This approach improved the binding of free water and sensory characteristics of the final product [3,4]. However, the possibility of replacing traditional stabilization systems containing hydrocolloids and chemically modified emulsifiers with natural polyfunctional ingredients—such as β-glucans—that possess foaming, emulsifying, and stabilizing properties, remains underexplored.

It is well known that the physicochemical properties of ice cream determine its thermodynamic stability [5]. The phenomenon of recrystallization, which occurs during storage of ice cream at low temperatures, leads to a gradual increase in the average size of ice crystals and a concomitant decrease in product quality [6]. To limit the excessive growth of ice crystals, proteins [7,8], polysaccharides [9], or their mixtures [10], and compositions of mono- and disaccharides [11,12] are used in various types of ice cream. Polysaccharides are considered to be the most commercially available and tested for cryoprotection in food [13]. The functions of carrageenan are among the most studied in ice cream technology [14,15,16]. However, interest in other polysaccharides is also growing. β-glucans are rapidly gaining popularity in the food industry, but their cryoprotective roles in frozen desserts remain largely unexamined. The functional and technological properties of β-glucans in ice cream depend on their origin, degree of purification, and interaction with recipe components [17,18]. Most studies in ice cream have focused on the use of oat and barley β-glucans, which increase the viscosity of ice cream mixes, resistance to melting, and limit ice crystal growth [19,20,21]. In this study, oat β-glucan was selected over barley due to its higher water-binding capacity, neutral sensory profile, wider regulatory acceptance, and well-documented rheological behavior in food systems. Despite these advantages, its functionality in acidophilic whey ice cream—particularly based on hydrolyzed demineralized whey—has not been studied.

It may be equally interesting to study the effect of yeast β-glucans on the dynamics of ice formation in ice cream. Most studies focus on the biological functions of yeast β-glucans, such as immunity enhancement, antioxidant capacity, and the inhibition of the growth of pathogenic microflora [22]. Tomczyńska-Mleko et al. [23] reported a synergistic effect found between β-glucan from yeast and κ-carrageenan from the stabilization system, leading to the formation of a stable gel network in whey mixes. However, their potential to prevent ice recrystallization and improve structural stability during frozen storage has not been previously evaluated.

Moreover, the literature contains contradictory interpretations of how polysaccharides influence ice crystallization and water mobility [24]. These effects depend on the type and concentration of the polysaccharide, as well as its molecular structure, processing conditions, and interaction with other ingredients.

Therefore, the aim of this study was to evaluate the potential of β-glucans from oats and yeast as natural stabilizers in whey-based ice cream formulated with a liquid concentrate of hydrolyzed demineralized whey. Specifically, the study sought to (1) assess the impact of these β-glucans on key physicochemical properties of the ice cream, including freezing point, overrun, and melting resistance; and (2) examine their effect on ice crystal size and recrystallization dynamics over one month of frozen storage, in comparison with a conventional stabilization system.

## 2. Materials and Methods

### 2.1. Raw Materials

Whey powder with the degree of demineralization of 90% (Herkules, MLEKOVITA, High Mazovia, Wysokie Mazowieckie, Poland), enzyme preparation lactase (β-D-galactosidase) with the activity of 5000 NLU/g (GODO-YNL2, Danisko, Brabrand, Denmark), activated starter *L. acidophilus* LYO 50 DCU-S (Danisko, Denmark) and water were used for the preparation of liquid hydrolyzed concentrate of demineralized whey with the solids content of 40%. Water, white sugar, vanillin, stabilization system Cremodan SI 320 (Danisco A/S, Brabrand, Denmark), and whey protein isolate 90% (SPOMLEK, Radzyń Podlaski, Poland) were used for ice cream production. Highly soluble β-glucan (1–3, 1–4) extracted from oats with a purity of 72% (Grupa Feniks 2050, Ćmielów, Poland) and β-glucan from yeast *Saccharomyces cerevisiae* with a purity of 70% (GOLDCELL, Biorigin, Sao Paulo, Brazil) were chosen as natural stabilizing ingredients.

The content of whey protein isolate (3%) was determined at the preliminary stage of the study as such that in combination with liquid hydrolyzed whey concentrate ensures the formation of satisfactory quality indicators [4]. The content of the stabilization system at 0.6% was used in accordance with the manufacturer’s recommendations for low-fat ice cream technology, and β-glucans were used in accordance with the available data in the scientific literature on their use in ice cream technology [19,20,23].

The formulations of the experimental ice cream samples are given in Table 1. 

### 2.2. Ice Cream Production

The starter was activated in ultra-pasteurized skimmed milk at 38–42 °C to pH 5.4–5.2. Liquid whey concentrate with a solids content of 40% was obtained by reconstitution of demineralized whey powder in water at 40–45 °C. The whey concentrate was filtered, pasteurized at 85–88 °C for 3–5 min, cooled to 40–43 °C, and β-D-galactosidase and starter *L. acidophilus* were added sequentially. Enzymolysis was carried out for 10 h until the degree of lactose hydrolysis was at least 95%. The technology of liquid hydrolyzed concentrates is described in the work of Mykhalevych et al. [25].

To prepare the ice cream samples, the dry components listed in Table 1 were first dispersed in water preheated to 40–45 °C under continuous stirring, followed by gradual incorporation of the liquid whey concentrate. The resulting mixture was filtered using a mesh with 1 mm pore size to remove any coarse particles. Pasteurization was performed at 83–87 °C for 5 min. Immediately afterward, the mixes were homogenized at a pressure of 12.0 ± 2.5 MPa using a 15M-8TA laboratory homogenizer and submicron disperser (Gaulin Corporation, Boston, MA, USA).

The homogenized mixes were cooled to 38–42 °C, after which 3% of an activated starter culture was added. Fermentation proceeded until the pH reached 5.25–5.10, after which the product was cooled to 2–6 °C, flavored with vanillin, and subjected to an aging phase lasting 12 h. Freezing was conducted in an FPM-3.5/380-50 “Elbrus-400” continuous freezer (JSC ROSS, Kharkiv, Ukraine). In the first phase of freezing, the mix was cooled to −1 °C in a 7 L cooling chamber at a scraper rotation speed of 4.5 s^−1^ for 120 s. In the second phase, simultaneous freezing and aeration were carried out at 9 s^−1^ for 180 s, reaching a final temperature of −5.0 ± 0.5 °C.

Each sample batch (3 kg per formulation) was hardened and stored at −18 ± 1 °C for 1 month using a Caravell A/S storage unit (Løgstrup, Denmark). To ensure reproducibility, three independent production runs were carried out for each ice cream formulation with identical composition.

### 2.3. Methods

#### 2.3.1. Chemical Composition and Physicochemical Properties

The solids content was determined by the arbitration method upon drying the samples at 105 °C [4], the protein content was determined by the Kjeldahl method [26], the fat content by the Gerber method [27], and the carbohydrate content by the Bertrand method [28].

The mass fraction of carbohydrates (lactose, glucose, and galactose) in whey concentrates was determined by the method of high-performance liquid chromatography on a chromatograph of model LC-6A (Shimadzu, Kyoto, Japan) with a refractometric detector, column SCR-101-N (250 × 4.7 mm). Deionized degassed water was used as an eluent, and the flow rate was 0.5 mL/min [25].

The freezing point was measured using a Marcel osm 3000 osmometer (Marcel, Waldenburg, Poland) with an accuracy of 0.002 °C. Each sample was equilibrated to room temperature before measurement, and approximately 50 µL of the mix was analyzed according to the manufacturer’s protocol [29].

The ice cream overrun (%) was determined by the weight method and calculated using the formula [30]O = (M_1_ − M_2_/M_2_) × 100,(1)
where M_1_ and M_2_ represent the mass of the same volume of unfrozen mix and frozen ice cream, respectively.

The resistance to melting (time of the first drop) was determined in samples of hardened ice cream (35 × 50 mm), which were placed on a grid (d = 95 mm, 5 × 5 mm holes, 0.5 mm wire thickness) and kept at an ambient temperature of 22 °C.

#### 2.3.2. Analysis of Ice Crystals

Ice cream samples were collected from at least three different locations within each specimen, ensuring a minimum depth of 3 cm from the surface to avoid surface artifacts. A small portion of each sample was transferred onto a glass microscope slide using a sterile spatula and carefully covered with a coverslip. To observe the recrystallization behavior of free water, the samples were examined under an Olympus BX53 optical microscope equipped with a Linkam LTS420 (Tokyo, Japan) temperature-controlled stage (operating range: −196 °C to +420 °C) and an Olympus SC50 digital imaging system (Tokyo, Japan). Image acquisition and analysis were performed using NIS Elements D software (v. 5.30.00, Nikon, Tokyo, Japan). For each condition, between 300 and 500 ice crystals were manually marked and analyzed for parameters including area, equivalent diameter, and standard deviation. The methodology is consistent with approaches previously reported in ice cream microstructure analysis [16,31].

#### 2.3.3. Statistical Processing

The frequency distribution of crystal size was calculated using macro data analysis in Microsoft Excel 2019. The relative frequency of each class interval was calculated as the number of crystals in that class (class frequency) divided by the total number of crystals and expressed as a percentage. The X50 parameter was analyzed as the average diameter (AD) for 50% of the crystals in the sample. All measurements were performed in triplicate on independently prepared batches (n = 3), and for each batch, a minimum of 100 ice crystals were measured to ensure statistical robustness. A completely randomized design was used for parallel sample processing. Analysis of variance (ANOVA) was performed using STATISTICA 13.1 software. The significance level was set at α = 0.05. Data are expressed as means with standard deviations (±SD), and differences between groups were evaluated using Tukey’s HSD test.

## 3. Results and Discussion

### 3.1. Chemical Composition and Physicochemical Parameters

The chemical composition of the experimental samples differs significantly from traditional whey ice cream (Table 2). Although whey-based ice cream typically contains a high proportion of free water and low levels of solids, fat, and protein—as noted in the Introduction—this limitation was addressed in this study by applying a previously developed formulation. Specifically, the use of liquid hydrolyzed demineralized whey concentrate in combination with whey protein isolate allowed us to produce low-fat whey ice cream with a solids content (42.05–42.61%) comparable to full-fat analogs (10–18% fat) [32]. The protein content (5.98–6.09%) also classifies the product as protein-enriched [33]. The carbohydrate component consisted of lactose (0.72–0.78%) and its hydrolysis products (32.30–32.39%), which may influence the freezing behavior of free water in the mix.

In the production of ice cream, controlling the freezing point is important to achieve the desired product texture [34]. In particular, this is essential in order to prevent the formation of large ice crystals. Structure stabilizers influence the freezing point indirectly by additionally binding free water, which leads to an increase in the solution concentration of low-molecular-weight compounds. In these circumstances, the three-dimensional grid is also reinforced, which disrupts the process of ice crystal formation by affecting the orderly arrangement of water molecules (Table 3).

The freezing point of the control sample of ice cream without stabilizers (C) indicates the natural behavior of ice cream when the mix is frozen. Under such conditions, ice crystals are able to form relatively freely, which can potentially lead to the formation of a coarse crystalline structure due to the formation of large ice crystals [35]. The SS sample containing 0.6% Cremodan SI 320 showed a positive trend in terms of freezing point reduction. Similar results have been reported in other studies examining the impact of commercial stabilization systems in ice cream [31,36]. Conversely, certain scientists posit that a decrease in the freezing point of up to 0.5 °C is insufficient to significantly inhibit recrystallization in ice cream [37]. This claim will be discussed in the following section on ice formation dynamics in ice cream.

The use of oat β-glucan (0.25–0.5%) has been demonstrated to reduce the freezing point most effectively, with an increase in the content of this polysaccharide resulting in a further enhancement of this effect. A slightly different trend is demonstrated by β-glucan from yeast, which has been observed to increase the freezing point compared to the control sample. Yeast-derived β-glucan is a polysaccharide with hydrophilic properties [38] that exhibits a somewhat distinct structure. Similarly to oat β-glucan, yeast-derived β-glucan interacts with water molecules to form hydrated networks, but β-glucans from cereals have an increased ability to bind free water [39].

However, the principal distinction in the behavior of yeast and oat β-glucan in ice cream is attributable to their molecular structure and water-binding properties. Yeast β-glucan is a high-molecular-weight branched polysaccharide that forms a dense gel network through hydrogen bonding, limiting water mobility and thereby inhibiting ice crystal growth despite raising the freezing point [40]. Oat β-glucan, being more linear and less branched, forms a viscous solution that more efficiently lowers the freezing point and stabilizes air cells [41]. Both types of β-glucans act through distinct mechanisms that contribute differently to the inhibition of recrystallization.

A particularly noteworthy finding is that yeast β-glucan, despite increasing the freezing point, effectively inhibits ice recrystallization during frozen storage. This apparent paradox can be explained by its structural mechanism of action. Yeast β-glucan consists of a β-(1→3)-linked backbone with frequent β-(1→6) side chains, resulting in a highly branched, non-linear structure. It forms a stable, highly branched gel matrix within the aqueous phase. This matrix physically entraps free water, significantly reducing molecular mobility and water diffusion. As a result, it limits the recrystallization process. Unlike colligative agents that depress freezing point through solute effects, yeast β-glucan stabilizes the system by creating a viscoelastic barrier against mass transfer. This structural immobilization of water offers a novel cryoprotective strategy, contrasting with the solute-driven action of oat β-glucan.

The freezing point of traditional ice creams typically ranges between −3.6 °C and −2.4 °C [42]. Consequently, it can be posited that the capability of yeast β-glucans to form stable gels in aqueous solutions also enhances their effectiveness in stabilizing ice cream, as demonstrated by the observed freezing point [43].

The composition of the ice cream also plays an important role in reducing the freezing point. The high content of solids (42.05–42.24%), in particular monosaccharides and protein (5.98–6.09%), also indirectly affects the reduction in the freezing point by binding free water. Scientists have noted that the presence of monosaccharides in ice cream typically results in a depression of the freezing point [44,45]. The hydrolysis products, monosaccharides (glucose and galactose), have a higher solubility than lactose [46]. This is due to the presence of numerous strongly polar hydroxyl groups in their molecules, which are capable of forming more hydrogen bonds than lactose [47,48]. Given that the monosaccharide content is significantly higher than in traditional ice cream, this also had an impact on the reduction in freezing point. Khaliduzzaman et al. [49] reported a freezing point range of −2.06 °C to −3.47 °C with an increase in solids from 35.58% to 36.42% and the simultaneous replacement of sugar with honey (up to 18%), which contains monosaccharides, specifically glucose and fructose. The obtained freezing point ranges indicate that these ice cream samples can be classified as food systems with a strong three-dimensional network, which is capable of resisting the formation of large ice crystals.

With regard to the resistance to melting and overrun of ice cream, it should be noted that the values in question are also influenced by the chemical composition of the ice cream, in particular the content of fat, proteins, and stabilizers [1]. Oat β-glucan’s higher solubility and capacity to create a more viscous and stable solution render it more effective in trapping and stabilizing air bubbles [50,51], resulting in an increase in overrun up to 81.52–83.12%. The formation of a more uniform texture and the stabilization of the ice cream structure may potentially slow down the melting process of ice cream. The viscosity of ice cream mixes with oat β-glucan and the degree of stabilization of the air phase indirectly affect the resistance to melting. Conversely, yeast β-glucan may not be as effective in stabilizing air bubbles, which results in slightly lower overrun values of 76.74–77.39%. Aljewicz et al. [19] reported that the use of 1% oat β-glucan resulted in an overrun of 73.45%, while Abdel-Haleem and Awad [52] found that 0.4% barley β-glucan decreased the overrun to 60.15%. The discrepancy in the outcomes can be attributed to the varying chemical compositions of ice cream, the degree of purification of additives, and the specific contents utilized in the study.

Additionally, β-glucan derived from yeast has been demonstrated to enhance the viscosity of ice cream mixes [53]. Nevertheless, its impact on resistance to melting is less pronounced in comparison to oat β-glucan. On the first day of storage, an increase in resistance to melting was observed for both samples containing oat β-glucan (from 24.01 min to 27.95–29.12 min) and the sample containing yeast β-glucan (from 24.26 min to 25.81 min). The incorporation of 0.25–0.5% of oat β-glucan and 0.5% of yeast β-glucan resulted in a more pronounced enhancement in the resistance to melting compared to the stabilization system.

During the storage period, all samples exhibited increased melting resistance. However, this effect was less pronounced in samples with yeast β-glucan compared to those with oat β-glucan. Yeast β-glucan did not engage the same level of interaction as oat β-glucan with other ice cream components that contribute to the melting resistance. In contrast to oat β-glucan, which is known for its pseudoplastic behavior in dairy systems and can result in excessive product density, yeast β-glucan forms a gel network with moderate strength [54,55].

### 3.2. Microscopy Analysis

The measurement of ice crystal size in the ice cream samples showed significant (*p* ≤ 0.05) differences in their growth pattern depending on the stabilizer used (Table 4). The control sample (C), which contained no stabilizing ingredients, showed an increase in ice crystal size from 18.50 μm on the first day to 27.50 μm after one month of storage (Table 4, Figure 1). This trend is consistent with the tendency of ice crystals to grow and coalesce over time in the absence of stabilizers, resulting in a coarser texture of frozen foods [56].

The SS sample containing 0.6% of the stabilization system initially had a smaller ice crystal size (15.80 μm after 24 h) than the control sample (Table 4). However, during storage there was a significant (*p* ≤ 0.05) increase in size to 32.15 μm (Table 4, Figure 1). This indicates that Cremodan SI 320 provides only initial stabilization and that its further efficacy decreases with prolonged storage. The SS sample showed greater crystal growth during storage than the control, suggesting that Cremodan SI 320 stabilizing effect is temporary and diminishes under prolonged cold storage due to weakening gel interactions. Although the scientific literature has reported the ability of commercial stabilizers and their mixes to provide a long-term effect in inhibiting the recrystallization of free water [57,58] some scientists consider that stabilizers do not significantly affect the initial crystal size distribution [59]. Similarly, stabilizers do not significantly affect ice crystal growth during freezing and hardening, but do affect ice recrystallization and may reduce the rate of ice crystal growth during cold storage. Of course, these properties can vary depending on the composition of the stabilization system, the mass fraction of the additive, and the component composition of the ice cream.

The incorporation of β-glucan derived from oats and yeast had a pronounced impact on the formation and growth of ice crystals in comparison to the control ice cream and the sample containing the stabilization system. The ice cream with 0.25% oat β-glucan exhibited an initial ice crystal size comparable to that of the control, yet during the storage period, the dynamics of their growth were observed to be slower, resulting in an average ice crystal size of 20.01 μm (Table 4). An increase in the concentration of oat β-glucan to 0.5% enhanced the inhibitory effect on ice crystal formation and significantly reduced their size, confirming its cryoprotective role. However, the highest stabilization effect (*p* ≤ 0.05) was observed when using β-glucan from yeast. In the sample with 0.25% yeast β-glucan, the smallest ice crystals were formed (8.49 μm), followed by a minimal increase to 9.52 μm after one month (Table 4, Figure 1). Similarly, the sample with 0.5% yeast β-glucan exhibited a slight increase in ice crystals from 10.24 μm to 11.08 μm over the same period (Table 4, Figure 1). The high efficiency of yeast β-glucan in inhibiting the growth of ice crystals can be explained by its ability to form a more stable gel network, which is able to more effectively limit water mobility and recrystallization due to the presence of branched (1–6) chains.

Due to their increased water binding capacity, β-glucans are able to capture more free-water molecules, which reduces the number of ice crystals formed and their further growth. Oat β-glucan forms a viscous solution that reduces the mobility of free water and the potential for large ice crystals to form. Yeast β-glucan forms a more stable gel network in the ice cream matrix and limits water mobility even more effectively. This stability is particularly useful for preventing the recrystallization of free water during long periods of ice cream storage.

The differences between oat and yeast β-glucans in their ability to inhibit ice crystal formation and growth in ice cream may also be due to the fact that partial degradation of β-glucan may occur during freezing due to a loss of solubility resulting from the formation of insoluble aggregates [60].

Low-temperature processing, such as repeated freezing and defrosting, can reduce the molecular weight, solubility and/or extractability of β-glucans from cereals. The reason is that freezing cannot inactivate the activity of β-glucanase enzymes responsible for breaking down β-glucan into low-molecular-weight fragments [61,62]. That is why temperature fluctuations during the production and storage of ice cream are undesirable for oat β-glucan, which can increase its destruction and, accordingly, reduce the ability to inhibit the recrystallization process. The behavior of β-glucan from cereals during freezing at low temperatures has been mainly studied in dough and bakery products [63,64], which is why further study of its mechanisms of action at low temperatures and during long-term storage of ice cream is a promising area.

The efficacy of β-glucan from yeast in limiting the growth of ice crystals is also subject to influence from processing, particularly pressure homogenization. Thammakiti et al. [65] investigated the impact of homogenization on the chemical composition, viscosity, and functional characteristics of β-glucan derived from the yeast *Saccharomyces cerevisiae*. The findings of this study indicated that the β-glucan preparation obtained following homogenization of yeast cells exhibited a higher β-glucan content and apparent viscosity. Homogenization was observed to result in cell wall fragmentation and enhanced release of β-glucan from yeast cells.

In the context of gel-forming properties of polysaccharides, it is important to recognize the pivotal role of the network structure of the gel in the effective stabilization of ice crystals. The gel-like network formed by oat β-glucan effectively retains water, but the stability of this network over time may be less robust than that of yeast β-glucan. The network formed by yeast β-glucan is more stable and resistant, thereby providing the greatest inhibition of ice crystal growth. It can be reasonably assumed that this network structure will prove more effective in maintaining smaller ice crystal sizes during long-term storage.

The analysis of photographs of ice crystals in ice cream during storage corroborates the identified patterns of free water recrystallization in the presence of various stabilizing agents (Figure 2). In the ice cream sample without stabilizers, the ice crystals formed on the first day are relatively small and evenly distributed (Figure 2A). However, after a week of storage, the ice crystals became larger and acquired an irregular shape (Figure 2B). After a month, free-water recrystallization and crystal growth into agglomerates occurred (Figure 2C). The stabilization system Cremodan SI 320 ensured the formation of evenly distributed ice crystals (Figure 2D), which remained relatively small during the first week of storage (Figure 2E). However, after one-month, significant growth (*p* ≤ 0.05) and agglomeration of ice crystals was observed (Figure 2F).

The incorporation of 0.25% oat β-glucan ensures the formation of uniform, small ice crystals (Figure 2G). These crystals exhibit a controlled tendency to increase and grow throughout the entire shelf life (Figure 2H,I). An increase in the content of oat β-glucan to 0.5% results in a more pronounced stabilization effect, as evidenced by the formation of smaller and more uniform ice crystals that impart a smooth texture to the product (Figure 2J–L). Nevertheless, the utilization of β-glucan derived from yeast (0.25–0.5%) ensures the formation of the smallest ice crystals and their uniform distribution within the ice cream matrix. Even after a period of one month, the crystals exhibit only a slight increase in size (Figure 2O,R). This substantiates the pronounced and long-term stabilizing effect of yeast β-glucan.

The scientific data presented in the article on the comparison of freezing point and patterns of free-water recrystallization in ice cream samples with different stabilizing ingredients expand the knowledge about the role of stabilizers in forming the texture and increasing the stability of ice cream during storage. A significant role of the type and amount of structure stabilizer on the recrystallization of free water in ice cream during storage for 1 month was demonstrated.

Understanding these results from a scientific point of view makes it possible to substantiate the formulation of ice cream to meet consumer expectations for a smooth and creamy texture. Further research is needed to investigate the mechanisms of action of β-glucans at low temperatures during longer shelf life of ice cream.

## 4. Conclusions

Stabilizing ingredients exhibit distinct effects on the physicochemical characteristics and ice crystal dynamics of whey-based ice cream. The commercial stabilization system Cremodan SI 320 (0.6%) reduced the freezing point to −2.74 ± 0.04 °C and maintained ice crystal size at 20.50 ± 0.77 μm after one week of storage, demonstrating short-term structural stability. Among natural alternatives, oat β-glucan showed the most pronounced effect on improving ice cream quality. At concentrations of 0.25–0.5%, it significantly reduced the freezing point (to −2.81 ± 0.05 °C), increased overrun (up to 42.5 ± 1.3%), and enhanced melting resistance. The 0.5% oat β-glucan sample displayed a fine initial ice crystal structure (X50 = 16.31 ± 0.15 μm), with moderate recrystallization after one month (20.01 ± 0.72 μm), suggesting effective short- to mid-term cryoprotection. Yeast β-glucan had a different mechanism of action. At 0.25–0.5%, it increased the freezing point (up to −2.62 ± 0.02 °C) and showed lower overrun and melting resistance compared to oat β-glucan. The 0.25% yeast β-glucan sample showed the smallest initial ice crystals (X50 = 8.49 ± 0.37 μm) and limited growth after one month (9.52 ± 0.16 μm), indicating effective long-term stabilization. In summary, oat β-glucan at 0.5% concentration offers the best improvement in overrun, freezing point depression, and short-term structural stability, while 0.25% yeast β-glucan is optimal for long-term inhibition of recrystallization during storage. These findings support the use of specific β-glucan types and concentrations as natural stabilizing agents in whey-based ice cream formulations.

## Figures and Tables

**Figure 1 foods-14-02184-f001:**
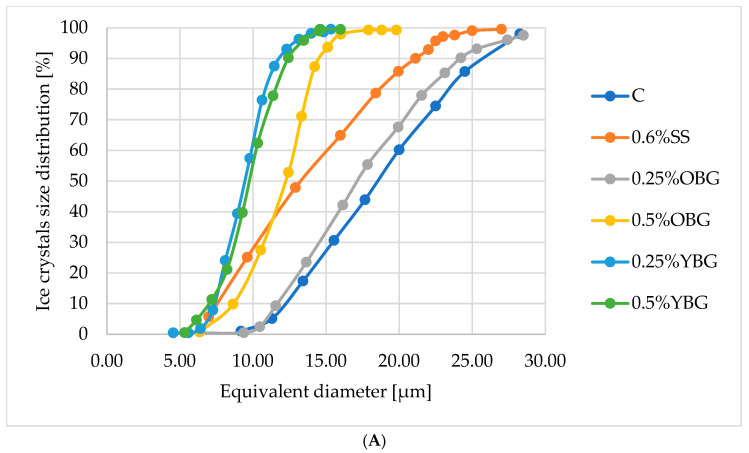
Distribution of ice crystals in whey ice cream samples after different storage periods: ((**A**) 24 h, (**B**) 1 week (1 W) and (**C**) 1 month (1 M)). C—control sample of ice cream without stabilizing ingredients; 0.6% SS—control sample of ice cream containing an additional 0.6% of the stabilization system Cremodan SI 320; 0.25% OBG—sample of ice cream containing 0. 25% β-glucan from oats; 0.5% OBG—ice cream sample containing 0.5% β-glucan from oats; 0.25% YBG—ice cream sample containing 0.25% β-glucan from yeast; 0.5% YBG—ice cream sample containing 0.5% β-glucan from yeast.

**Figure 2 foods-14-02184-f002:**
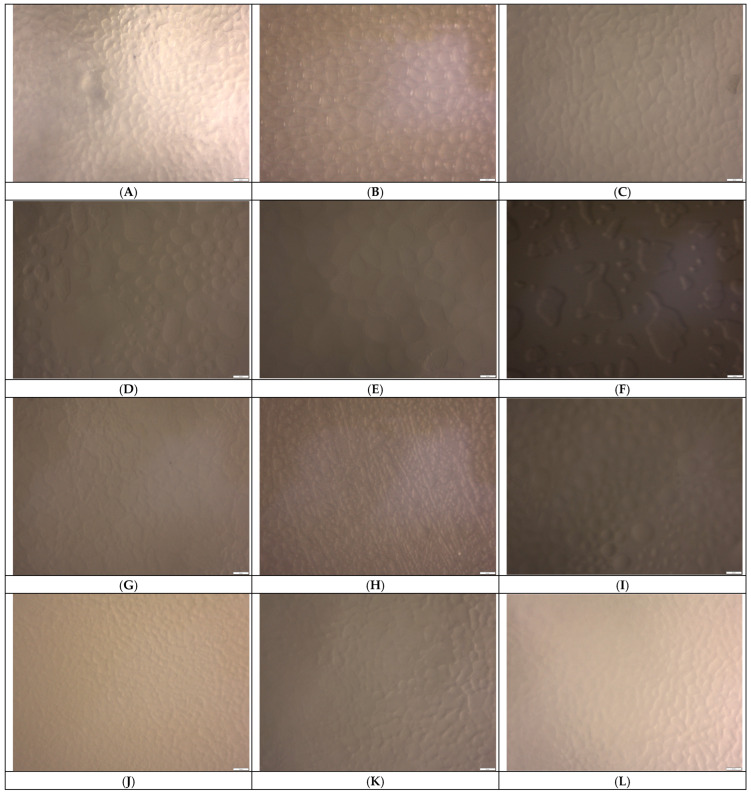
Photographs of ice crystals in whey ice cream samples after different storage periods (24 h (**A**,**D**,**G**,**J**,**M**,**P**), 1 W (**B**,**E**,**H**,**K**,**N**,**Q**), 1 M (**C**,**F**,**I**,**L**,**O**,**R**): (**A**–**C**) control ice cream sample without stabilizing ingredients; (**D**–**F**) control ice cream sample containing 0.6% of the stabilization system Cremodan SI 320; (**G**–**I**) ice cream sample containing 0.25% of β-glucan from oats; (**J**–**L**) ice cream sample containing 0.5% β-glucan from oats; (**M**–**O**) ice cream sample containing 0.25% β-glucan from yeast; (**P**–**R**) ice cream sample containing 0.5% β-glucan from yeast.

**Table 1 foods-14-02184-t001:** Formulations of the experimental samples of whey ice cream.

Ingredients, %	Labeling of Ice Cream Samples					
	C	0.6% SS	0.25% OBG	0.5% OBG	0.25% YBG	0.5% YBG
Liquid hydrolyzed concentrate of demineralized whey	75.0	75.0	75.0	75.0	75.0	75.0
White sugar	9.0	9.0	9.0	9.0	9.0	9.0
Whey protein isolate	3.0	3.0	3.0	3.0	3.0	3.0
Stabilization system Cremodan SI 320 (Danisco A/S, Brabrand, Denmark)	–	0.6	–	–	–	–
β-glucan from oats	–	–	0.25	0.5	–	–
β-glucan from yeast	–	–	–	–	0.25	0.5
Activated starter	3.0	3.0	3.0	3.0	3.0	3.0
Vanillin	0.1	0.1	0.1	0.1	0.1	0.1
Water	9.9	9.3	9.65	9.4	9.65	9.4
Total	100.0	100.0	100.0	100.0	100.0	100.0

Note: C—control sample of ice cream without stabilizing ingredients; 0.6% SS—control sample of ice cream containing an additional 0.6% of stabilization system Cremodan SI 320; 0.25% OBG—sample of ice cream containing 0.25% of β-glucan from oats; 0.5% OBG—ice cream sample containing 0.5% β-glucan from oats; 0.25% YBG—ice cream sample containing 0.25% β-glucan from yeast; 0.5% YBG—ice cream sample containing 0.5% β-glucan from yeast.

**Table 2 foods-14-02184-t002:** Chemical composition of ice cream.

Sample	Solids, %	Protein, %	Fat, %	Carbohydrates, %	Lactose, %	Monosaccharides, %
C	42.05 ^a^ ± 0.91	6.05 ^a^ ± 0.05	0.32 ^a^ ± 0.01	33.08 ^a^ ± 0.95	0.73 ^a^ ± 0.05	32.35 ^a^ ± 0.09
0.6% SS	42.61 ^a^ ± 1.02	6.01 ^a^ ± 0.02	0.73 ^e^ ± 0.02	33.05 ^a^ ± 1.24	0.75 ^a^ ± 0.02	32.30 ^a^ ± 0.11
0.25% OBG	42.28 ^a^ ± 0.67	6.09 ^a^ ± 0.06	0.37 ^c^ ± 0.01	33.11 ^a^ ± 1.00	0.72 ^a^ ± 0.05	32.39 ^a^ ± 0.05
0.5% OBG	42.50 ^a^ ± 0.95	6.03 ^a^ ± 0.01	0.40 ^d^ ± 0.02	33.12 ^a^ ± 0.98	0.75 ^a^ ± 0.03	32.37 ^a^ ± 0.32
0.25% YBG	42.33 ^a^ ± 1.12	5.98 ^a^ ± 0.05	0.35 ^b^ ± 0.01	33.14 ^a^ ± 1.05	0.78 ^b^ ± 0.01	32.36 ^a^ ± 0.10
0.5% YBG	42.24 ^a^ ± 1.04	6.01 ^a^ ± 0.03	0.39 ^d^ ± 0.01	33.09 ^a^ ± 1.32	0.76 ^a^ ± 0.06	32.33 ^a^ ± 0.13

Note: C—control sample of ice cream without stabilizing ingredients; 0.6% SS—control sample of ice cream containing an additional 0.6% of stabilization system Cremodan SI 320; 0.25% OBG—sample of ice cream containing 0.25% of β-glucan from oats; 0.5% OBG—ice cream sample containing 0.5% β-glucan from oats; 0.25% YBG—ice cream sample containing 0.25% β-glucan from yeast; 0.5% YBG—ice cream sample containing 0.5% β-glucan from yeast. ^a–e^—mean values denoted in columns by different letters differ statistically significantly at *p* ≤ 0.05.

**Table 3 foods-14-02184-t003:** Physicochemical characteristics of mixes and ice cream.

Sample	Freezing Point, °C	Overrun, %	Resistance to Melting, Min		
			24 h	1 W	1 M
C	−4.222 ^b^ ± 0.14	69.08 ^a^ ± 2.55	24.01 ^ab^ ± 1.15	24.32 ^a^ ± 0.64	24.58 ^a^ ± 0.23
0.6% SS	−4.688 ^b^ ± 0.03	75.25 ^b^ ± 1.82	25.42 ^b^ ± 0.57	26.12 ^b^ ± 0.44	27.03 ^b^ ± 0.80
0.25% OBG	−5.108 ^c^ ± 0.25	81.52 ^d^ ± 3.24	27.95 ^c^ ± 0.46	28.53 ^c^ ± 0.97	29.96 ^c^ ± 1.09
0.5% OBG	−6.040 ^d^ ± 0.18	83.12 ^d^ ± 2.61	29.12 ^d^ ± 0.84	29.87 ^c^ ± 0.65	31.05 ^d^ ± 1.37
0.25% YBG	−3.888 ^a^ ± 0.07	77.39 ^b^ ± 2.48	24.26 ^a^ ± 0.53	24.58 ^a^ ± 0.38	25.61 ^a^ ± 0.94
0.5% YBG	−3.846 ^a^ ± 0.02 ^a^	76.74 ^b^ ± 2.04	25.81 ^ab^ ± 1.20	26.07 ^b^ ± 0.31	26.50 ^b^ ± 0.72

Note: C—control sample of ice cream without stabilizing ingredients; 0.6% SS—control sample of ice cream containing an additional 0.6% of stabilization system Cremodan SI 320; 0.25% OBG—sample of ice cream containing 0.25% of β-glucan from oats; 0.5% OBG—ice cream sample containing 0.5% β-glucan from oats; 0.25% YBG—ice cream sample containing 0.25% β-glucan from yeast; 0.5% YBG—ice cream sample containing 0.5% β-glucan from yeast; 24 h—24 h of ice cream storage, 1 W—1 week of ice cream storage, 1 M—1 month of ice cream storage. ^a–d^—mean values denoted in columns by different letters differ statistically significantly at *p* ≤ 0.05.

**Table 4 foods-14-02184-t004:** Dynamics of ice crystal growth in ice cream during 1 month of storage.

Sample	Time of Storage	Minimum Diameter of Ice Crystals (μm)	Maximum Diameter of Ice Crystals (μm)	Average Value of Ice Crystal Diameter (μm)
C	24 h	9.18 ^a^ ± 0.12	28.26 ^a^ ± 0.70	18.50 ^a^ ± 1.21
	1 W	12.23 ^b^ ± 0.03	35.37 ^b^ ± 0.89	25.01 ^b^ ± 1.06
	1 M	13.33 ^c^ ± 0.14	37.39 ^c^ ± 0.52	27.50 ^c^ ± 0.78
0.6% SS	24 h	5.64 ^a^ ± 0.22	30.32 ^a^ ± 0.46	15.80 ^a^ ± 0.67
	1 W	10.60 ^b^ ± 0.11	35.71 ^b^ ± 0.35	20.50 ^b^ ± 0.77
	1 M	16.72 ^c^ ± 0.47	43.84 ^c^ ± 0.60	32.15 ^c^ ± 1.18
0.25% OBG	24 h	5.32 ^a^ ± 0.12	28.31 ^a^ ± 0.42	18.74 ^a^ ± 0.04
	1 W	7.27 ^b^ ± 0.02	30.07 ^b^ ± 0.65	19.29 ^b^ ± 0.50
	1 M	8.03 ^c^ ± 0.05	36.60 ^c^ ± 1.05	20.01 ^b^ ± 0.72
0.5% OBG	24 h	6.35 ^a^ ± 0.19	19.81 ^a^ ± 0.28	11.38 ^a^ ± 0.17
	1 W	8.35 ^b^ ± 0.16	27.59 ^b^ ± 0.89	12.71 ^b^ ± 0.16
	1 M	9.52 ^c^ ± 0.12	30.55 ^c^ ± 0.71	16.31 ^c^ ± 0.15
0.25% YBG	24 h	4.54 ^a^ ± 0.03	15.33 ^a^ ± 0.41	8.49 ^a^ ± 0.37
	1 W	4.68 ^b^ ± 0.02	16.51 ^a^ ± 0.64	9.26 ^b^ ± 0.12
	1 M	4.73 ^b^ ± 0.04	17.19 ^b^ ± 0.31	9.52 ^b^ ± 0.16
0.5% YBG	24 h	5.32 ^a^ ± 0.19	15.99 ^a^ ± 0.50	10.24 ^a^ ± 0.02
	1 W	5.45 ^a^ ± 0.09	19.31 ^b^ ± 0.98	10.52 ^a^ ± 0.49
	1 M	7.08 ^b^ ± 0.18	20.72 ^c^ ± 0.52	11.08 ^b^ ± 0.20

Note: C—control sample of ice cream without stabilizing ingredients; 0.6% SS—control sample of ice cream containing an additional 0.6% of stabilization system Cremodan SI 320; 0.25% OBG—sample of ice cream containing 0.25% of β-glucan from oats; 0.5% OBG—ice cream sample containing 0.5% β-glucan from oats; 0.25% YBG—ice cream sample containing 0.25% β-glucan from yeast; 0.5% YBG—ice cream sample containing 0.5% β-glucan from yeast; 24 h—24 h of ice cream storage, 1 W—1 week of ice cream storage, 1 M—1 month of ice cream storage. ^a–c^—mean values denoted (according to storage time within the group) in the columns by different letters are statistically significantly different at *p* ≤ 0.05.

## Data Availability

The original contributions presented in the study are included in the article, further inquiries can be directed to the corresponding author.

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
