# Peer review of "The Effect of β-Glucans from Oats and Yeasts on the Dynamics of Ice Crystal Growth in Acidophilic Ice Cream Based on Liquid Hydrolyzed Whey Concentrate"

_foods, 2025, doi:10.3390/foods14132184_

Round 1

Reviewer 1 Report

Comments and Suggestions for Authors

This manuscript investigates the effects of β-glucans from oats and yeasts on the quality indicators and freezing dynamics of whey ice cream based on liquid hydrolyzed demineralized whey concentrate. The topic is novel and holds practical relevance, particularly in the exploration of natural stabilizing ingredients. The experimental design covers key ice cream quality parameters and ice crystal analysis. However, the manuscript exhibits several significant issues regarding consistency of results, logical flow of arguments, clarity of expression, and formatting standards, which require a major revision.

Abstract

The terms “stabilization system” (Lines 20, 26) and “stabilizing system” (Line 35) are used interchangeably. It is recommended to use one consistent term throughout the manuscript.

The abstract fails to effectively highlight the core innovation of the study. Please enhance the impact of the study by clearly identifying the unique advantages of the two β-glucans investigated compared to traditional stabilizers (or findings from other studies) to enhance the study’s impact.

Introduction

Line 68 The authors mention both oat and barley β-glucans but the study only investigates β-glucan from oats without explaining the rationale behind the choice. The scientific basis for selecting oat β-glucan over other sources should be clearly stated.

Line 83-88 The statement of research objectives should be more explicitly stated at the end of the paragraph.

Materials and Methods

Line 145-146 Please give the reference.

Line 149 Please give more details about the freezing point.

Line 168 The temperature “-420 °C” is physically impossible. Please verify and correct the actual operating temperature range.

Line 175-182 The Statistical processing section should specify the design of the parallel experiment to ensure the reliability of the results.

Line 178 The abbreviation of “average diameter” should be “AD”.

Results and Discussion

Line 185-192 Please verify if the analysis of the whey ice cream's chemical composition presented here is consistent with the information provided in the Introduction (Lines 41-45).

Line 202 The subsection number “3.2” appears to be missing here. Please check and correct the numbering.

Line 211 The space between symbols and values appears and disappears from time to time. It is best to unify the whole manuscript. Please check the whole manuscript.

Line 245-246 It is stated here that yeast β-glucan lowers the freezing point. This directly contradicts the descriptions in the abstract (Line 24), the results discussion (Lines 233-235), and Table 3, which indicate it increases the freezing point. This is a significant contradiction that must be clarified and unified throughout the manuscript.

Line 249-250 Similarly, it is stated here that oat β-glucan increases the freezing point, whereas the abstract (Lines 22-23), results discussion (Lines 231-232), and Table 3 indicate it lowers the freezing point. Please verify and unify this statement.

Line 239-254 The discussion regarding the relationship between the structure and performance of oat and yeast β-glucan requires more robust and accurate scientific basis.

Line 296-297 The description of the effect of yeast β-glucan on melting resistance here is inconsistent with the trend described in the Abstract (Lines 24-26) during storage. Please verify the data and unify the conclusions.

Line 311 This should be Table 4, not the previously referenced Table 3. Please correct the table number here and check all cross-references throughout the text.

Line 324-325 It would be beneficial to further explain why the ice crystal size of the SS sample even exceeded that of the control group during storage.

Line 338-340 The description here regarding the inhibitory effect of 0.5% oat β-glucan on ice crystal formation and growth contradicts the claim in Lines 249-252 that oat β-glucan promotes the formation of larger ice crystals. Please verify.

Line 357 The “0.6%CC” should be “0.6%SS”.

The ice crystal size distribution shown in Figure 1 does not seem to match data in Table 3 (e.g., a maximum diameter of 43.84 ± 0.60 μm). Please carefully check the raw data against the chart presentation to ensure consistency.

Line 376 The term should be “β-glucanase”.

Figure 2 can be more clear

The apparent paradox that yeast β-glucan increases the freezing point yet effectively inhibits ice crystal growth is a potentially key finding of this paper. However, the discussion fails to provide a convincing mechanistic explanation. This is a major flaw that the authors must address by substantially strengthening this part of the discussion.

Conclusions

The conclusions are too general and lack specificity, failing to fully reflect the depth and specific findings of the study. Please revise and improve them. For instance, the conclusions should clearly state the specific effect differences observed among groups with different concentrations of beta-glucan, and potentially identify an optimal stabilizer concentration, rather than just providing a broad summary.

Reviewer 2 Report

Comments and Suggestions for Authors

Dear Authors,

I have attached my comments and suggestions to improve the manuscript further. Otherwise, the study is very interesting

Reviewer 3 Report

Comments and Suggestions for Authors

The manuscript investigates the impact of oat and yeast β-glucans on the physicochemical properties and ice crystal growth dynamics in whey-based ice cream made from liquid hydrolyzed whey concentrate. The study provides a detailed analysis of how stabilizing ingredients, including a commercial stabilizer (Cremodan SI 320) and β-glucans, affect freezing point, overrun, resistance to melting, and ice crystal size during one month of storage. The authors demonstrate that oat β-glucans significantly lower the freezing point and enhance overrun and melting resistance, while yeast β-glucans excel in inhibiting ice crystal growth, contributing to a stable gel network. The work is well-structured, scientifically sound, and relevant to the field of food technology, particularly for low-fat ice cream formulations. Its strengths include a robust experimental design, clear data presentation, and a focus on natural stabilizers, which aligns with current industry trends.

GENERAL CONCEPT COMMENTS

The methodology is generally sound, with detailed descriptions of raw materials, production processes, and analytical techniques, allowing for reproducibility. While a few analytical methods would benefit from appropriate referencing, the overall experimental approach is robust. The discussion is mostly well-structured and supported by experimental data, although several statements require clarification or correction for consistency and scientific accuracy. The reference list is relevant and mostly up to date, but minor formatting inconsistencies and a few citation errors should be addressed.

SPECIFIC COMMENTS:

Line 5-6 (Author List Inconsistency): There is an inconsistency between the list of authors provided in the manuscript PDF and the Review Report Form. Specifically:

  • In the PDF, the listed authors are: Artur Mykhalevych, Galyna Polishchuk, Agata Znamirowska-Piotrowska, Anna KamiÅ„ska-Dwórznicka, Maciej Kluz, and Magdalena Buniowska-Olejnik.
  • In the Review Report Form, Anna KamiÅ„ska-Dwórznicka is missing, and Maciej Kluz appears twice—once under the full name Maciej Ireneusz Kluz and again as Maciej Kluz.

It is recommended that the authors verify and unify the authorship information across all submission documents to ensure consistency and avoid confusion during the editorial process.

Line 34 (Keywords): The list currently includes 10 keywords may be too extensive for standard indexing purposes and could be refined for conciseness. Please consider limiting this to 6–8. Suggested removals: "overrun" (already included under “quality indicators”), "microscopy" (covered by “ice crystals”), and "stabilization system" (or replace with Cremodan SI 320). Suggested keyword list: ice cream, liquid whey concentrate, freezing point, ice crystals, quality indicators, oat β-glucan, yeast β-glucan.

Line 110 (Table 1): Please clarify that the “Stabilization system” refers specifically to Cremodan SI 320, as readers unfamiliar with commercial stabilizers may find this ambiguous.

Line 145-146 (Methods – Chemical Composition): Analytical methods for compositional parameters (protein, fat, lactose, etc.) are described without reference to validated protocols. Please cite standard references (e.g., AOAC, IDF, or ISO methods) to substantiate the techniques used and enhance reproducibility.

Line 156 (Methods – Overrun): The formula is appropriate; however, for clarity, please specify that M1 and M2 represent the mass of the same volume of unfrozen mix and frozen ice cream, respectively. This avoids potential misinterpretation by readers.

Line 241-252 (Discussion – β-glucan Mechanisms): Several issues merit attention:

  • Contradictory Statements: The manuscript states that oat β-glucan reduces the freezing point (as shown in Table 3), but later suggests it may increase it due to particle aggregation. These claims are contradictory and not clearly supported by references [37, 38]. Please revise to distinguish speculative mechanisms from experimental results, and remove unsupported conclusions.
  • Structure Description: Yeast β-glucan is described as a “linear polymer” yet also “highly branched.” This is contradictory. A more accurate phrasing would be: “Yeast β-glucan consists of a β-(1→3)-linked backbone with frequent β-(1→6) side chains, resulting in a highly branched, non-linear structure.”
  • Unsupported Claims: The assertion that oat β-glucan may act as nucleation sites to increase the freezing point (line 248) is speculative and not substantiated by either references or data. Consider removing or clearly labeling this statement as a hypothesis requiring further research.

Line 285 (Discussion – Overrun): The reference to Abdel-Haleem and Awad [49] incorrectly states that barley β-glucan increased overrun to 60.15%. Please verify the original study, which suggests a decrease in overrun. This inconsistency should be corrected and, if relevant, the discrepancy with current findings discussed.

Line 311 (Table Reference): The reference to “Table 3” in the context of ice crystal growth data should be “Table 4” to align with the correct table numbering. Please update the table numbering throughout the manuscript for consistency.

Line 422-436 (Conclusions): The conclusions are generally supported by the data, but the statement that yeast β-glucan’s ability to form a stable gel network “compensates” for its higher freezing point could be more precisely explained. Please elaborate on how this compensation occurs or rephrase for clarity.

References (Line 466-609): There are inconsistencies in numbering and sequencing. For instance, reference [56] appears to be missing, and Thammakiti et al. is cited as [60] in the main text but listed as [63] in the reference list. Please ensure that all in-text citations match the bibliography and that numbering is sequential.

Round 2

Reviewer 1 Report

Comments and Suggestions for Authors

Although the authors have made some changes according to the requirements of the reviewers, the changes in Table 4 are still confusing. After the caption of Table 4 was modified, Table 3 was still used in the text. The table referenced in the text must be correct. Line 328, Line 330, Line 345-346, Line 364, Line 369-Line 370.

Author Response

Comment: Although the authors have made some changes according to the requirements of the reviewers, the changes in Table 4 are still confusing. After the caption of Table 4 was modified, Table 3 was still used in the text. The table referenced in the text must be correct. Line 328, Line 330, Line 345-346, Line 364, Line 369-Line 370.

Response: Thank you for support and evaluation of the manuscript. We corrected the mistakes (higlighted in pink throughout the text).